# Dynamics-dependent density distribution in active suspensions

Jochen Arlt [1], Vincent A. Martinez [1], Angela Dawson[1], Teuta Pilizota [2] & Wilson C.K. Poon[1]

Self-propelled colloids constitute an important class of intrinsically non-equilibrium matter. Typically, such a particle moves ballistically at short times, but eventually changes its orientation, and displays random-walk behaviour in the long-time limit. Theory predicts that if the velocity of non-interacting swimmers varies spatially in 1D, $v(x)$, then their density $\rho(x)$ satisfies $\rho(x) = \rho(0)v(0)/v(x)$, where $x = 0$ is an arbitrary reference point. Such a dependence of steady-state $\rho(x)$ on the particle dynamics, which was the qualitative basis of recent work demonstrating how to 'paint' with bacteria, is forbidden in thermal equilibrium. Here we verify this prediction quantitatively by constructing bacteria that swim with an intensity-dependent speed when illuminated and implementing spatially-resolved differential dynamic microscopy (sDDM) for quantitative analysis over millimeter length scales. Applying a spatial light pattern therefore creates a speed profile, along which we find that, indeed, $\rho(x)v(x) =$ constant, provided that steady state is reached.

[1] School of Physics & Astronomy, The University of Edinburgh, Peter Guthrie Tait Road, Edinburgh EH9 3FD, UK. [2] School of Biological Sciences and Centre for Synthetic and Systems Biology, The University of Edinburgh, Alexander Crum Brown Road, Edinburgh EH9 3FF, UK. Correspondence and requests for materials should be addressed to J.A. (email: j.arlt@ed.ac.uk) or to V.A.M. (email: vincent.martinez@ed.ac.uk)

Einstein predicted and Perrin verified that, in a gravitational field, the equilibrium (number) density of a dilute dispersion of colloidal particles varied with height $z$ according to

$$\rho(z) = \rho(0)e^{-z/z_0}, \qquad (1)$$

where $z_0$ encodes the equality of diffusive and sedimentation fluxes:

$$z_0 = \frac{D_0}{v_s}, \qquad (2)$$

with $v_s$ a particle's sedimentation speed and $D_0$ its thermal diffusivity. For spheres of radius $a$ in a liquid of viscosity $\eta_0$ at temperature $T$ with density $\Delta$ lower that of the particles $v_s = 2ga^2\Delta/9\eta_0$ and $D_0 = k_BT/6\pi\eta_0 a$ where $k_B$ is Boltzmann's constant, so that $z_0 = 3k_BT/4\pi ga^3\Delta$. The verification of Eq. (1) demonstrated the granularity of matter[1].

Now suppose $\eta_0 = \eta_0(z)$, so that the particle dynamics also varies in space: $D_0 = D_0(z)$. Nevertheless, statistical mechanics stipulates that the spatially dependent dynamical coefficient $D_0(z)$ cannot appear in the equilibrium density distribution, and Eq. (1) still holds because the $\eta$-dependence cancels out in Eq. (2).

Active colloids[2], particles that dissipate energy to propel themselves, form an important class of active matter[3,4]. Such dissipative states of matter, which include all living organisms, are intrinsically non-equilibrium, and give rise to new physics. Consider a system of run-and-tumble particles (RTPs). An RTP self propels ('runs') at velocity $\mathbf{v}$ for time $\tau_{run}$, then changes direction ('tumbles') instantaneously to run at $\mathbf{v}'$ such that $|\mathbf{v}'| = |\mathbf{v}|$ but with randomised direction, so that at times $\gg \tau_{run}$ it behaves as a random walker. Suppose the run speed of such particles is spatially dependent, $v(\mathbf{r})$. Solving a Fokker-Planck equation for the coarse-grained kinetics of RTPs, Tailleur and Cates[5,6] predicted that the resulting density distribution should be $\rho(\mathbf{r}) = \rho(0)v(0)/v(\mathbf{r})$, with $\mathbf{0}$ an arbitrary origin. A purely mechanical derivation is also possible[7]. The appearance of the particles' dynamics, $v(\mathbf{r})$, in this formula contrasts starkly with the sedimentation equilibrium of passive colloids with spatially dependent diffusivity $D_0(x)$, for which the $D$-independent Eq. (1) still holds.

When restricted to 1D, this result becomes

$$\rho(x) = \rho(0)v(0)/v(x), \qquad (3)$$

which was first derived for non-interacting random walkers by Schnitzer[8]. Later, Tailleur and Cates showed that it is valid for interacting RTPs whose run speed can be expressed as $v(x) = v[\rho(x)]$[5]. Moreover, under quite general conditions, $\rho v = $ constant also holds (provided translational diffusion is negligible) for active Brownian particles (ABPs)[9], which reorient gradually due to their rotational diffusivity $D_{rot}$, losing directional memory after a persistence time of $\sim D_{rot}^{-1}$. At $t \gg D_{rot}^{-1}$, an ABP is again a random walker.

Qualitatively, Eq. (3) was the basis of recent demonstrations of templated self assembly using light-activated motile bacteria[10,11]. In a spatially varying illumination pattern, cells accumulate in the darker regions, generating contrast. Quantitatively, however, Eq. (3) has remained unverified by experiments to date. Moreover, its theoretical derivations do not include hydrodynamic interactions, so that its applicability to real systems is open to doubt.

Here, we investigate Eq. (3) quantitatively with the same light-activated *Escherichia coli* bacteria used previously to demonstrate templated active self assembly[10]. Our strain's rapid response to changes in light intensity allows us to impose well defined changes in local swimming speed. Differential dynamic microscopy (DDM)[12] has previously been shown to reliably measure swimming speed and density of motile bacteria on uniform samples[10,13]. Here we develop and demonstrate a spatially resolved implementation (sDDM) for quantitative analysis of $\bar{v}(x, y)$ and $\rho(x, y)$ over larger length scales ($\sim$mm) with a resolution of $\approx$100 μm. We use sDDM to confirm that indeed $\rho(x)v(x) = $ constant, provided that steady state is reached.

## Results

**Light controlled *E. coli* and spatially resolved analysis.** Each *E. coli* is an $\approx 2 \times 1$ μm spherocylinder with $\approx 7$–$10$ μm helical flagella powered by rotary motors[14]. When all flagella rotate counter-clockwise (seen from behind), they bundle and propel the cell. Every $\tau_{run} \sim 1$ s or so, one or more flagella reverse and unbundle, causing a change in direction: wild-type (WT) cells are RTPs[15]. At a typical average speed $\bar{v} \gtrsim 10$ μm s$^{-1}$, they random walk with a persistence length $l_p \sim \bar{v}\tau_{run} \sim 10$ μm. Deleting the *cheY* gene prevents tumbling; cells become ABPs with $D_{rot}^{-1} \sim 10$ s and $l_p \sim \bar{v}D_{rot}^{-1} \sim 100$ μm[16], so that at times $\gg D_{rot}^{-1}$ cells random walk with $D_{eff} \sim \bar{v}^2 D_{rot}^{-1} \sim 10^3$ μm$^2$ s$^{-1}$.

Our *E. coli* mutants carried a plasmid expressing proteorhodopsin (PR), which pumps protons when exposed to green light[17]. Cells suspended in nutrient-free motility buffer were sealed into 20 μm high compartments and imaged using 10× phase contrast microscopy. After some minutes, $\bar{v}$ dropped abruptly to zero upon oxygen exhaustion[10]. Thereafter, cells only swam when illuminated in green[10,18,19], with an average speed $\bar{v}$ that increased with the light intensity, $\mathcal{I}$, saturating at $v_{max}$ beyond some $\mathcal{I}_0$[10]. These are living analogues of synthetic light-activated active colloids[20,21].

We first used a digital mirror device[10] to project a binary (bright-dark) spatial intensity pattern $\mathcal{I}(x, y)$ spelling out 'UoE' (inset of Fig. 1a) onto a field of cells that had been uniformly illuminated for some time, so $\bar{v}$ was initially constant in space. We used sDDM to measure $\bar{v}(x, y)$, $\beta(x, y)$ and $\rho(x, y)$ in 64 × 64 (pixel)$^2$ tiles (see Methods section and Supplementary Note 1 for details). The projected $\mathcal{I}(x, y)$ was rapidly replicated in a pattern of $\bar{v}(x, y)$ (Fig. 1a). A similar $\rho(x, y)$ pattern soon forms (Fig. 1b) so that this effect can be used for templated self assembly[10,11]. Given that higher cell densities occur in darker regions with lower swimming speed, Eq. (3) is clearly qualitatively correct[10,11]. We now proceed to test it quantitatively.

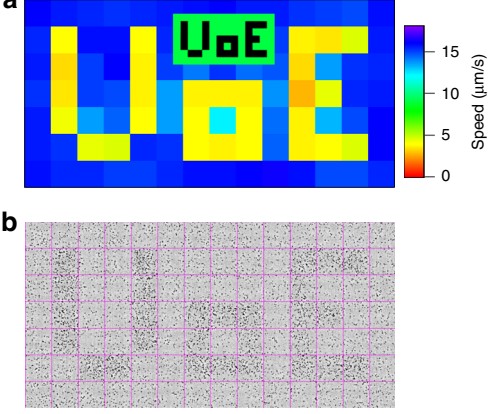

**Fig. 1** Demonstration of spatial-resolved differential dynamic microscopy. Projecting light patterns onto a sample of light-powered *E. coli* AD10 (at OD = 1) leads to a spatial variation of the mean swimming speeds $\bar{v}$. **a** Map of $\bar{v}$ as measured by sDDM when the negative 'UoE' pattern shown in inset is projected onto the sample. **b** Phase contrast image about 20 s after applying the pattern, showing accumulation in the darker regions. Magenta lines indicate the square tiles of 64 pixels (90 μm) size used for the sDDM analysis

We projected a quasi-1D stepped illumination pattern

$$\mathcal{I}(x,y) = \begin{cases} \mathcal{I}_+ < \mathcal{I}_0, & x < 0 \\ \mathcal{I}_- < \mathcal{I}_+, & x > 0 \end{cases} \qquad (4)$$

on a field of these cells. This generated a spatial pattern of swimming speed, $\bar{v}(x,y)$, and cell density, $\rho(x,y)$, which we quantified using sDDM. Averaging over $y$ gives $\bar{v}(x)$ and $\rho(x)$, which allows a direct test of Eq. (3), provided that this light pattern (Eq. (4)) generates a corresponding sharp pattern of cell run speeds:

$$\bar{v}(x) = \begin{cases} \bar{v}_+, & x < 0 \\ \bar{v}_- < \bar{v}_+, & x > 0 \end{cases}. \qquad (5)$$

This requires cells that respond rapidly to changes in $\mathcal{I}$, which was found not to be the case[10] for previous PR-expressing mutants with otherwise intact metabolism[18]. Indeed, a recent attempt to verify Eq. (3) using PR-bearing *E. coli* found instead (in our notation) $\rho = (a/\bar{v}) + b$ with positive constants $a$ and $b$. The latter was ascribed to a long $\tau_{stop}$, which led to memory effects[11].

We achieved rapid response by deleting the *unc* gene cluster encoding the $F_1F_o$-ATPase membrane protein complex from a parent K12-derived $\Delta cheY$ mutant, giving a fast-responding smooth-swimmer, AD10[10]. In fully-oxygenated phosphate motility buffer (MB) at optical density OD $= 1$, $\bar{v} \approx 30~\mu m~s^{-1}$ and a fraction $\beta \lesssim 20\%$ of cells were non-motile. When illuminated anaerobically, $\bar{v}_{max} = 28~\mu m~s^{-1}$ and $\tau_{stop} \ll 1$ s, compared with a $\tau_{stop}$ of many minutes in the parent strain without *unc* deletion[10]. (Details of other strains we used are given in the methods section.)

Strictly speaking, a non-interacting limit does not exist for bacterial suspensions[22]. Cells interact hydrodynamically at any concentration, although simulations show that swimmers behave effectively as non-interacting when $\rho/\rho_c \lesssim 0.1$, where $\rho_c$ is the density for the onset of collective behaviour. We observed collective motion in our *E. coli* suspension at OD $\gtrsim 10$, corresponding to a cell body volume fraction of $\phi \gtrsim 1.4\%$, consistent with a previous estimate of 2%[22,23], so that a quasi-non-interacting limit is reached at OD $\lesssim 1$. It was not possible to work below this limit because of an increasing fraction of cells trapped in circular trajectories (due to hydrodynamic interactions with walls of the sample chamber[24]) that did not explore the whole sample compartment, hindering relaxation towards a steady state. We therefore worked at OD $\geq 1$. We report first data for OD $= 5$ ($\rho \approx 5 \times 10^9$ cells/ml; $\phi \approx 0.7\%$) before discussing OD $= 1$, where the data are noisier due to lower cell numbers.

**Stepped light pattern at OD $= 5$.** A field of AD10 cells rendered stationary by oxygen exhaustion was uniformly illuminated for $\approx 20$ min to achieve saturation speed[10]. The light was then attenuated to $\mathcal{I}_-$, the level of the darker half of the target pattern (Eq. (4)) for 5 min to determine $\bar{v}_- = 6.5(2)~\mu m~s^{-1}$. Returning the intensity to its initial level, we waited another 5 min for the swimming speed to return to $\bar{v}_+ = 13.2(2)~\mu m~s^{-1}$. We measured the cell density $\rho_0$ and non-motile fraction $\beta_0$ of this high-speed uniform sample, and then switched on a stepped pattern (Eq. (4)) by reducing the intensity in the $x > 0$ half plane.

Figure 2a shows the mean swimming speed averaged over $y$ tiles, $\langle \bar{v}(x,y) \rangle_y = \bar{v}(x)$, normalised to the whole-sample-averaged speed, $\langle \bar{v} \rangle$, plotted against $x$ at 30 min after switching on the stepped pattern. A stepped speed pattern was developed (Fig. 2a, black circles).

If Eq. (3) is valid, we expect the swimmer density to obey

$$\rho_+^s(x)\bar{v}_+(x) = \rho_-^s(x)\bar{v}_-(x), \qquad (6)$$

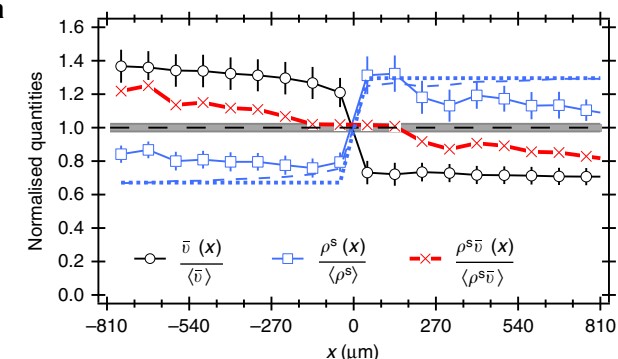

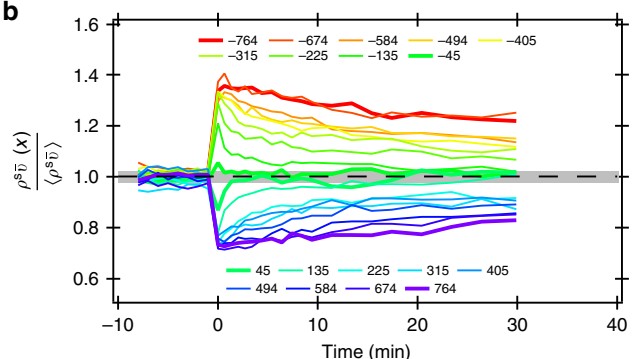

**Fig. 2** Response of AD10 at OD $= 5.3$ to a stepped intensity pattern. **a** Normalised quantities as indicated vs. $x$ after $\approx 30$ min of stepped illumination, and predictions for normalised $\rho^s$ based on two speeds (dotted line) and complete measured $\bar{v}(x)$ profile (dashed line). Error bars are s.e.m. of the $y$-averaged values. **b** Time dependence of normalised $\rho^s \bar{v}$ at various $x$ (see legend). The sample was uniformly illuminated before the halves pattern was switched on at $t = 0$. In both plots the grey area corresponds to the standard deviation in experimental values for uniform illumination ($t < 0$)

where '±' subscripts having their obvious meanings. If the density of non-motile cells is constant throughout the experiment (see Supplementary Note 2 for justification), i.e.

$$\rho_\pm^{nm} = \beta_0 \rho_0, \qquad (7)$$

we can write the total cell density on the two sides of $x = 0$ as

$$\rho_\pm = \rho_\pm^s + \rho_\pm^{nm} = \rho_\pm^s + \beta_0 \rho_0. \qquad (8)$$

Finally, the average cell density is

$$\rho_0 = \frac{1}{2}(\rho_+ + \rho_-). \qquad (9)$$

Equations (6)–(9) together predict the density of motile cells in the two half-planes:

$$\rho_\pm^s = \frac{2\rho_0(1 - \beta_0)}{1 + \frac{v_+}{v_\mp}}. \qquad (10)$$

We calculated the swimmer density in our experiments from the measured total cell density $\rho(x)$ and non-motile fraction $\beta(x)$ using $\rho^s(x) = \rho(x)[1 - \beta(x)]$, and normalised it by the whole-sample-averaged swimmer density. This function 30 min after the imposition of the stepped intensity pattern is also stepped (Fig. 2a, blue squares), with the theoretical predictions from Eq. (10), using the measured average $v_\pm$ as inputs (Fig. 2a, dotted line), giving a reasonable account of the step amplitude. A more sophisticated version of this model which takes the measured shape of $\bar{v}(x)$ into account (see Supplementary Note 2 for details)

is able to capture the amplitude of the jump in $\rho^s(x)$ at $x = 0$ even more precisely (Fig. 2a, dashed line).

The product $\rho^s \bar{v}(x)$ normalised to the whole sample average (Fig. 2a, red x's), is indeed constant for $|x| \lesssim 200\,\mu m$, verifying Eq. (6), which is the application of Eq. (3) to our conditions. However, there are systematic deviations from constancy at $|x| \gtrsim 200\,\mu m$. One possible explanation is the emergence of collective motion with associated local nematic ordering[25], which would invalidate the derivation of Eq. (3). However, we only observed collective motion at OD $\gg 10$. Instead, the deviation of $\rho^s \bar{v}$ from constancy at $|x| \gtrsim 200\,\mu m$ is a kinetic effect. Figure 2b shows the time evolution of the normalised $\rho^s \bar{v}(x)$ at different $x$. Steady state was reached rapidly for $|x| \lesssim 200\,\mu m$, but was not reached by 30 min at the extremes of our observation window, $|x| \gtrsim 600\,\mu m$. Given their effective diffusivity $D_{\mathrm{eff}} \sim 10^3\,\mu m^2\,s^{-1}$, cells at the extremities of our compartment take $\gg 30$ min to sufficiently sample both speed regions, preventing the attainment of steady state within our observational time window. This leads to the deviations between observed and predicted $\rho^s(x)$ away from $x = 0$. Nevertheless, Fig. 2b suggests that $\rho^s \bar{v} = $ constant should be attained at all $x$ in the long-time limit.

**Stepped pattern at other cell densities**. Measurements and model predictions for the lower OD $= 1$ are shown in Fig. 3. The data are noisier, but show the same trends. In the vicinity of $x = 0$, $\rho^s \bar{v} \approx$ constant. To highlight the behaviour in the two 90 $\mu m$-wide stripes of tiles bordering $x = 0$, we plot $\rho^s \bar{v}(x)$ at $t = 30$ min for these two stripes against each other for a number of independent experiments (Fig. 4, black solid circles). In all cases, $\rho^s \bar{v}(x) =$ constant for these central stripes to within experimental uncertainties. The ratio of $\rho^s \bar{v}(x)$ in these two stripes plotted against the ratio of the swimming speed on the two half-planes (Fig. 4 inset, black solid circles) is consistent with this claim.

We performed experiments using the stepped light pattern at other cell densities and also using an additional smooth swimming strain (DM1). In all cases up to OD $= 8$, we find that $\rho^s \bar{v}(x)$ is constant across the two central stripe of tiles on either side of $x = 0$ (Fig. 4), where we are certain that a steady state has been reached, verifying Eq. (3) up to $\rho \approx 8 \times 10^9$ cells/ml. Spatial maps and time evolution of spatial profiles for our highest density sample (OD $= 8$) are shown in Supplementary Figs. 6 and 7, respectively.

**Measurements using a periodic light pattern**. The complicating factor so far is the slow global convergence towards $\rho^s(x)v(x) =$ const, so that steady state will only be reached in approximately hours. Experiments on such time scales are impractical due to mechanical and biological stability issues. Thus, we only have direct evidence for the validity of Eq. (3) in the vicinity of the intensity step at $x = 0$. This suggests that the use of a series of thin stripes would give more clear-cut results unencumbered by kinetic issues. We found that this was indeed the case.

In response to the imposition of a one-dimensional square-wave illumination pattern of brighter-darker stripes with 540 $\mu m$ repeat generated by a digital mirror device[10], the swimming speed of bacteria changed from a uniform distribution to a square-wave distribution almost instantaneously (in $\lesssim 1$ s) (Fig. 5a). This in turn modified the cell density (initially uniform at OD $= 1$), which approached a steady state much more quickly. This is possible not only because of the length scale reduction, but also because swimmers can enter (say) a high-intensity region from low-intensity regions on both the left and right.

Figure 5b shows the normalised swimmer density $\rho^s(x)/\langle \rho^s \rangle$ after 15 min of patterned illumination, together with $\bar{v}(x)/\langle v \rangle$ and their product. While the data are again somewhat noisy because

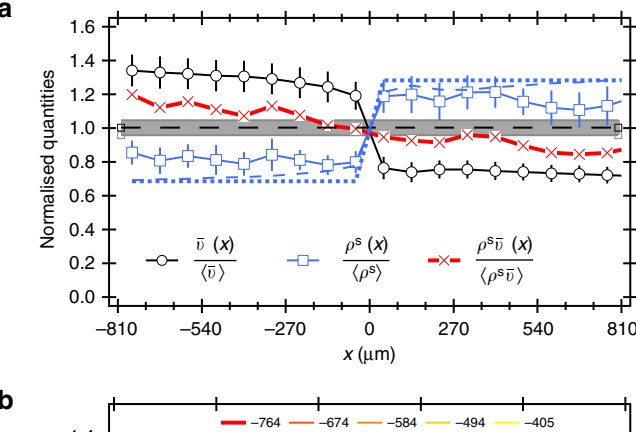

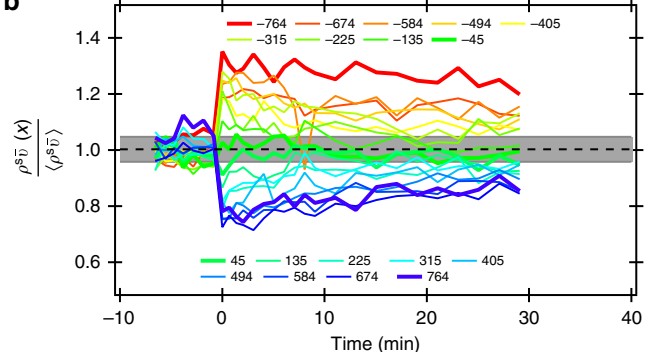

**Fig. 3** Response of AD10 at OD $= 1$ to a stepped intensity pattern. The same quantities as in Fig. 2 are shown. Qualitatively the behaviour is the same as for the higher density, but the data are noisier due to the overall weaker signal. See Supplementary Figs. 4 and 5 for spatial maps and time evolutions, respectively

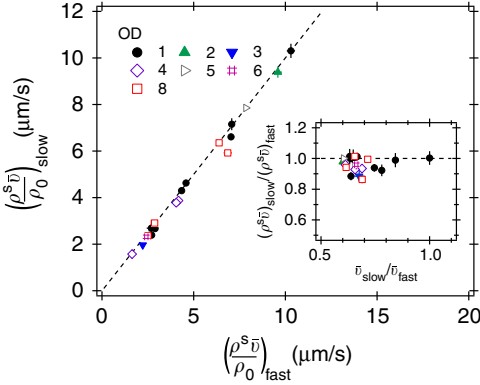

**Fig. 4** $\rho^s \bar{v}$ either side of an intensity step for various sample densities and speed ratios. The main plot shows $\rho^s \bar{v}$ for the first tile on the slow (-) side vs. the same quantity for the first tile on the fast (+) side for several independent datasets. For both low density (filled symbols) as well as higher-density datasets (open symbols) Eq. (6) holds. This is also demonstrated in the inset, which shows $(\rho^s_- \bar{v}_- / (\rho^s_+ \bar{v}_+))$ vs. $\bar{v}_- / \bar{v}_+$ for the same datasets. Error bars show s.d.

of the low average cell density (OD $= 1$), it is clear that $\rho^s \bar{v}(x) =$ constant to within one standard deviation, which directly verifies Eq. (3).

**Experiments with $\beta(I)$ dependency**. Interestingly, experiments using low light intensities (which gave low swimming speeds) proved less successful, because at very low intensities we found a noticeably higher percentage of non-motile cells in the sample

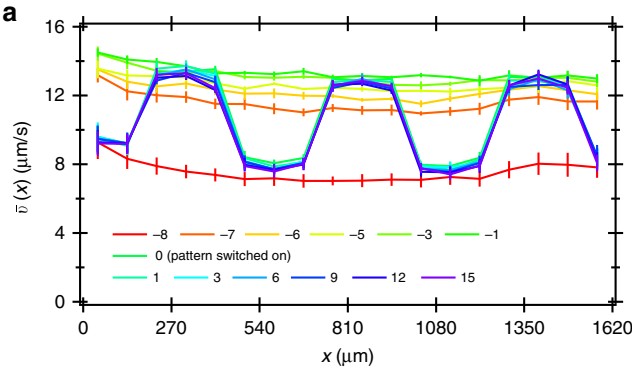

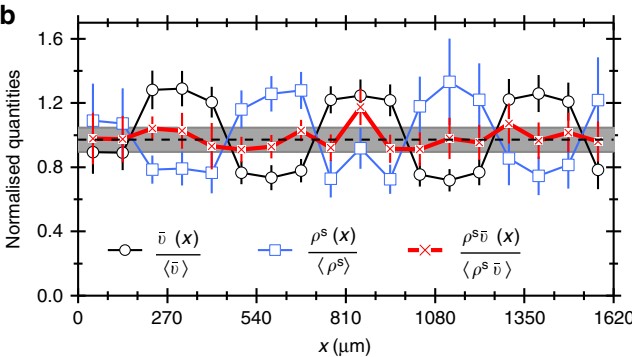

**Fig. 5** Response of bacteria to the imposition of a 1D square wave with 270 μm wide stripes. **a** The evolution of the speed profile normalised to its (time) average, with time increasing from red to violet (legend gives $t$ in minutes): initially illumination is uniform, with the intensity getting increased from low to high at $t = -7$ min, then the pattern is applied at $t = 0$ for 15 min. **b** The spatial profiles of various normalised quantities as indicated in the legend after 15 min of patterned illumination. Error bars represent s.d. Dashed line (and grey area) is the average ($\pm 1$ s.d.) over all tiles for the uniform case ($t < 0$). Error bars show propagated s.e.m.

than at high light intensities (Supplementary Fig. 8). This led to a spatial variation in the non-motile density (Supplementary Fig. 9), which considerably complicates the interpretation and analysis of such experiments (see Supplementary Note 3 for details).

**Experiments using light-powered run-and-tumble strain.** We end by explaining why we did not use motility wild-type (run and tumble) strains for our experiments. Their motion randomises much more rapidly than smooth swimming mutants, which would have significantly alleviated the non-steady-state issue for the stepped intensity pattern. However, we found that AD4, a PR-bearing motility wild type, gathered near $x = 0$, on the darker side of the intensity step (see Supplementary Figs. 10 and 11 and Supplementary Note 4 for details). From the $q$-dependence of our DDM data[13] we can deduce that the tumbling rate increases noticeably as cells swim from light to dark, whereas cells swimming from dark to light do not show any obvious change in their tumbling behaviour. This may be due to 'energy taxis'[26]. The validity of Eq. (3) depends on the assumption that the tumbling rate is independent of swimming direction[5,8], so that motility wild types cannot be used to test this result.

## Discussion

Equation (3) is one of only a handful of exact predictions to date on the statistical mechanics of active particle systems. Its 'weak' form, for non-interacting systems was derived for RTPs[8], while its

'strong' form was later derived both for RTPs[5] and ABPs[9]. Taken together, our experiments using stepped and stripped light patterns give strong evidence that Eq. (3) holds at $1 \leq OD \leq 8$ ($0.15\% \lesssim \phi \lesssim 1.2\%$) for smooth swimming *E. coli* whenever we can be sure that steady state has been reached, either in the vicinity of $x = 0$ in the stepped pattern or throughout the stripped pattern. Our swimmers are interacting throughout our concentration range[22], even though collective motion is not observed until $OD \gg 10$. Thus, our results verify the 'strong' form of Eq. (3) for ABPs.

The qualitative validity of Eq. (3), viz., that cells gather where they swim slower, or, equivalently for our cells, where the light intensity is lower, has already been assumed and utilised in recent work deploying such cells in smart (or reconfigurable) templated self assembly, or 'painting with bacteria'[10,11]. Indeed, in a recent demonstration of how to perform bacterial painting with multiple shades of graded intensity levels[11], there was attempt at checking the correctness of Eq. (3) en passant, which, however, was unsuccessful because of a high number of non-motile cells and the long stopping time of their strain, the latter producing memory effects. Our success in verifying Eq. (3) shows that carefully quantifying and subtracting the non-motile fraction and the use of a strain of bacteria with very short stopping time are essential ingredients in such an experiment. Indeed, without careful design most 'real' active systems are likely to display dynamic behaviour that is too complex to fulfil the assumptions leading to Eq. (3), as evidenced by our findings for the wild type strain with illumination-dependent tumbling rate. Note, however, that 'smart templated active self assembly'[10] using photo-activated swimmers is possible for any relationship of $\rho - v$ in which the increase in one variable necessitates the decrease in the other.

Our experiments would not have been possible without spatially resolved differential dynamic microscopy (sDDM), which can reliably quantify swimming speed and relative density (along with many other parameters) over a wide range of length scales and cell concentrations. Its adoption can therefore provide new insights into a variety of systems displaying spatially varying dynamics, from biological taxis[27] to collective motion.

Throughout, we have focussed on steady-state effects, although the consideration of time dependence proved crucial in interpreting apparent systematic deviations from the prediction of Eq. (3) for imposed stepped intensity patterns. Time-dependent effects are, of course, interesting in their own right. Thus, the response of active particles to a time-dependent topographic landscape that is self-assembled by the cells themselves[9] has yet to be explored experimentally. On the other hand, it has recently been suggested theoretically[28] and demonstrated experimentally[29] that travelling-wave light fields can be exploited for transporting and rectifying light-activated swimmers. Exploitation of these and other opportunities should open up new fields of fundamental studies and applications.

## Methods

**Strains and sample preparation.** We constructed three different strains of *E. coli* using plasmids expressing SAR86 $\gamma$-proteorhodopsin (a gift from Jan Liphardt, UC Berkley). These strains were designed to exhibit a fast response to changes in light intensity. This was achieved by deleting the *unc* gene cluster, so that the $F_1F_0$-ATPase membrane protein complex cannot work in reverse in the dark to generate proton motive force to power swimming. The detailed molecular biology and strain characterisation have been reported before[10]. AD4 is a WT (run-and-tumble) swimmer derived from AB1157, whereas DM1 and AD10 are smooth swimming strains derived from RP437 and AB1157, respectively (see Supplementary Table 1). The two smooth swimming strains behaved similarly, although AD10 achieved a much higher swimming speed than DM1 and was also more efficiently powered by light. Therefore we mostly used AD10, with some additional data acquired using DM1.

Overnight cultures were grown aerobically in 10 mL Luria-Bertani Broth (LB) using an orbital shaker at 30 °C and 200 rpm. A fresh culture was inoculated as 1:100 dilution of overnight grown cells in 35 ml tryptone broth (TB) and grown for 4 h to an optical density of $OD_{600} \approx 0.2$. The production of proteorhodopsin (PR) was induced by adding arabinose to a concentration of 1 mM as well as the necessary cofactor all-trans-retinal to 10 μM to the growth medium. Cells were incubated under the same conditions for a further hour to allow protein expression to take place and then transferred to motility buffer (MB, pH = 7.0, 6.2 mM $K_2HPO_4$, 3.8 mM $KH_2PO_4$, 67 mM NaCl and 0.1 mM EDTA). Single filtration (0.45 μm HATF filter, Millipore) was used to prepare high density stock solutions (OD ≈ 8) which were diluted with MB to the desired cell concentration.

The samples were loaded into commercial 2 μL sample chambers (SC-20-01-08-B, Leja, NL) of dimensions ≈ 6 × 10 mm × 20 μm, where cells predominantly swim in the $(x, y)$ (imaging) plane, but have enough room to 'overtake' each other in all three spatial dimensions. The chamber was then sealed using vaseline to stop air flow, so that swimming stopped once dissolved oxygen was exhausted[19]. This happened within 10 min at OD ≈ 1 (≈$10^9$ cells/ml or 0.2% volume fraction of cell bodies). Thereafter, we controlled the activity of the cells by illuminating with green light of various intensities[10].

**Experimental setup**. The samples were observed using a Nikon TE2000 inverted microscope with a PF 10×, N.A. 0.3 phase contrast objective. Time series of movies (~40 s duration at 100 frames per second) were recorded using a CMOS camera (MC 1362, Mikrotron). A long-pass filter (RG630, Schott Glass) in the bright-field light path ensured that the imaging light did not activate PR. The light controlling bacterial swimming was provided by an LED (Sola SE II, Lumencor) whose intensity was set via a computer interface. The LED light was filtered to a green wavelength range (510–560 nm) overlapping with the absorption peak of our PR[18] and illuminated the sample in a trans-illumination geometry. By illuminating an area much larger than the field of view of the objective, we minimised the loss of swimmers over time. If only a small region of the sample is illuminated, the density of swimmers continuously drops, because they reach the illumination boundaries and accumulate there (no light = no swimming). For the stepped pattern experiment we uniformly illuminated a ≈7 mm diameter circle, covering almost all of the sample chamber. Under these conditions the cell density is conserved, thus simplifying theoretical modelling. We used a thin sheet polariser imaged onto the sample plane to attenuate the intensity on half of the sample. A digital mirror device[10] projected the periodic pattern onto a ≈2.9 mm diameter area of the sample.

**Differential dynamic microscopy**. DDM measures $(\bar{v}, \beta)$ averaged over $10^4$–$10^5$ cells under our conditions[13,19,30]. From ≈40 s of wide-field, low-magnification movies, one extracts the power spectrum of the difference between pairs of images delayed by time $\tau$, $g(\vec{q}, \tau)$, where $\vec{q}$ is the spatial frequency vector. Under suitable conditions and for isotropic motion, the intermediate scattering function $f(q, \tau)$, the $q$th mode of the density autocorrelation function, is given by

$$g(q, \tau) = A(q)[1 - f(q, \tau)] + B(q). \qquad (11)$$

Here, $B(q)$ relates to the background noise and $A(q)$ is the signal amplitude. Fitting $f(q, \tau)$ to a suitable swimming model of E. coli yields four key motility parameters: the mean $\bar{v}(q)$, and width $\sigma(q)$ of the speed distribution $P(v)$, the non-motile fraction $\beta(q)$, and the diffusion coefficient of non-motile cells $D(q)$, as a function of $q$. All of these should, ideally, be $q$-independent. In practice, there is some $q$-variation. We typically averaged the fitting parameters over $0.5 < q < 2.2 \, \mu m^{-1}$ to give, e.g. $\bar{v} = \langle v(q) \rangle_q$ and $\beta = \langle \beta(q) \rangle_q$.

In a dilute system whose structure factor $S(q) \approx 1$, $A(q)$ is proportional to the sample density[31,32] and can therefore be used to determine relative densities by $\rho_1/\rho_0 = \langle A_1(q)/A_0(q) \rangle_q$[10]. Note that ratioing the $A(q)$s removes their strong $q$-dependence.

Spatially resolved DDM (sDDM) is in principle straightforward: the above algorithm simply needs to be implemented on $p \times p$ (pixel)$^2$ sub-movies. In practice, care is required in choosing the minimum $p$ for which meaningful results can be obtained. We do this by illuminating a field of cells uniformly, measuring $(\bar{v}, \beta, \rho, \rho^s v)$ from individual $p \times p$ tiles in the steady state, and obtaining the probability distribution of these parameters. Under our imaging conditions, we found that these distributions became $p$-independent when $p \geq 64$. We therefore chose $p = 64$, corresponding to 90 μm in the sample (see Supplementary Note 1 for details).

A full $512 \times 512$ movie yields $g(q, \tau)$s at $512/2 = 256$ distinct $q$ values. We divide it into 64 sub-movies of size $64 \times 64$ (pixel)$^2$. This yields $8 \times 8 \times (64/2) = 2048$ $g(q, \tau)$s to be fitted to give for each sub-movie $v_{x,y}(q)$, $\beta_{x,y}(q)$ and $\rho_{x,y}(q)/\rho_0 = A(q)_{x,y}/A_0(q)$, where $A_0(q)$ is measured from the same sample under uniform illumination (i.e. just before switching to a structured light pattern). These were averaged over $0.5 \leq q \leq 1.5 \, \mu m^{-1}$. The upper $q$ limit is somewhat lower than what is typical for whole-movie analysis[13] due to non-systematic failure of fitting at higher $q$ values, presumably due to noise or windowing artefacts[33].

## Data availability
The data presented here is available on the Edinburgh DataShare repository[34].

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

## Acknowledgements
The authors were funded by the EPSRC (EP/J007404/1) and the ERC (AdG 340877 PHYSAPS). We thank Mike Cates, Julien Tailleur, Alexander Morozov and Nick Koumakis for helpful discussions, Jan Liphardt for a gift of PR plasmids and Dario Miroli for *E. coli* strain DM1.

## Author contributions
J.A. and V.A.M. contributed equally to this work. W.C.K.P. initiated the work. T.P. and A.D. designed mutants constructed by A.D. J.A. and V.A.M. performed experiments, analysed and interpreted data with T.P. and W.C.K.P., and wrote manuscript with W.C.K.P.

## Additional information

**Competing interests:** The authors declare no competing interests.

