## [Peer Review File · Nature Communications]

Reviewer #1 (Remarks to the Author):

The paper by Arlt et. al. presents a systematic check of a theoretical prediction regarding the steady-state density of particles (in this case bacteria) whose speed is spatially dependent. The prediction states that the density should be inversely proportional to the local velocity. In its simplest form, it was first made by MJ Schnitzer (PRE 48, 2553 1993). A more elaborate form, where the velocity is a function of the local density, was put forward by Cates and Tailleur. Given the result by Schnitzer I suggest that the authors cite his paper.

The applicability of the results to the experiment is in no way obvious. This is in particular since the theory does not account for hydrodynamic interactions which are present in the real system. While the surface screens these interactions, the equivalence of the results is not completely obvious. The authors might like to mention this fact as opposed to quoting the theory as a fundamental one.

The paper is very well written and the results convincing. While some evidence of this result was already given in Ref 11 of the manuscript (Fig. 3) the current manuscript goes well beyond this. This was done by using both more systematic measurements and a strain of bacteria whose speed adapts quickly to a change in light. In this respect, I find the paper a significant contribution and recommend its publication. I suggest that the authors account for the comments given above.

Reviewer #2 (Remarks to the Author):

Currently, there is a growing interest in self-propelled systems in various forms, such as active colloids or autonomous robots, all more or less inspired by biological and social systems. Because they are out of equilibrium, novel theories must be developed to deal with such systems. This requires considerable effort to understand them fundamentally. The dependence of the steady-state particle density on their dynamical properties, as discussed by the authors, is a rather central prediction in this context, and it is definitely worth conducting extensive experimental testing.

Such a test is the main objective of the paper. As motile organisms, the authors use a mutant of E-Coli (developed in the Liphard Lab), which shows a rather immediate velocity response to changing conditions of light illumination. If the tested equation is true, these velocity changes should be reflected in their local density distribution. The authors provide experimental data for different patterns of light which partially agrees with theoretical predictions.

I think this is a well written and interesting paper which is certainly applicable for Nat. Comm. (despite quite some overlap with their previous Nat. Comm. publication). However, there are a couple of issues which should be addressed before I can support publication.

1. What is the reason for a circular E. coli motion at low concentrations?

2. The authors rationalize the deviations of the red symbols in Fig. 2a from a constant by saying that beyond $x = \pm 200 \mu\text{m}$ the steady state could not be reached. But I am less convinced about the data in Fig. 3a where an almost constant downward slope of the red symbols is observed over the entire range. Then the agreement with a horizontal confidence interval is not really convincing.

3. The authors also used stripe patterns (Fig. 5) with widths of $270 \mu\text{m}$. This is about 3 times the persistent length of the bacteria with the knocked out cheY gene. Is this really sufficient to reach a steady state? I would like to see some further evidence for this, as this is a crucial requirement for the tested relation?

4. Finally, it would be helpful to indicate some further implications of the tested relationship between particle density and particle dynamics. What are the consequences? How does it contribute to our understanding of collective behavior?

Reviewer #3 (Remarks to the Author):

In this article, the authors probe the theoretical (1D) prediction relating the spatial density of active particles to their spatially dependent speed. In analogy to their previous work published in Nature Communications and the work from Di Leonardo's group published in e-life, they use bacteria that swim with an intensity dependent speed under spatially variable illumination generated with a DMD.

There are three major shortcomings that prevent publication of this work in Nature Communications:

1. This effect is not surprising or new: the fact that density is inversely proportional to speed is known since the 90's and was, for example, reported by Schnitzer for chemotactic systems. Another way to see this, is that particles will concentrate where their diffusivity is lower, and this has been confirmed experimentally both with passive and active systems, e.g. with active baths.

2. This demonstration does not have the broad appeal and interest for a multidisciplinary community as claimed in the article. The same effect was assumed and exploited, at least qualitatively, by the authors in their previous work. Although, it's nice to see, in principle, an actual

quantification of this effect, I'm not convinced of its broad interest and applicability, especially when limited to a 1D case as here. Real-life problems are barely 1D but, at least, 2D. Moreover, these results have very limited applicability as they can't be used for example with wild-type phototactic microorganisms.

3. The data are not convincing or presenting strong evidence. In Fig. 3 and 5 the authors for example claim that the red line is constant within the experimental error. However, if one considers the surrounding trends, it is hard to believe how such a claim can be made with confidence: in Fig. 3 data could easily fall on a line and in Fig. 5 on a sinusoidal function. Finally, there is also the fact that, from a technical point of view, this work does not add much to what the authors have already published in their previous paper.

Response to reviewer's comments

Reviewer #1:

The paper by Arlt et. al. presents a systematic check of a theoretical prediction regarding the steady-state density of particles (in this case bacteria) whose speed is spatially dependent. The prediction states that the density should be inversely proportional to the local velocity. In its simplest form, it was first made by MJ Schnitzer (PRE 48, 2553 1993). A more elaborate form, where the velocity is a function of the local density, was put forward by Cates and Tailleur. Given the result by Schnitzer I suggest that the authors cite his paper.

Our manuscript did already cite MJ Schnitzer (PRE 48, 2553 1993) as its reference [8], but our wording of the relevant section was indeed unclear, possibly giving the impression ref. 8 was referring to work by Cates and Tailleur. We have updated the wording to make it unambiguous (bottom of p.2).

The applicability of the results to the experiment is in no way obvious. This is in particular since the theory does not account for hydrodynamic interactions which are present in the real system. While the surface screens these interactions, the equivalence of the results is not completely obvious. The authors might like to mention this fact as opposed to quoting the theory as a fundamental one.

We indeed agree that the applicability of the theoretical results to our experiment is not a foregone conclusion and added a sentence to stress this in the introductory section (page 3).

The paper is very well written and the results convincing. While some evidence of this result was already given in Ref 11 of the manuscript (Fig. 3) the current manuscript goes well beyond this. This was done by using both more systematic measurements and a strain of bacteria whose speed adapts quickly to a change in light. In this respect, I find the paper a significant contribution and recommend its publication. I suggest that the authors account for the comments given above.

Reviewer #2:

Currently, there is a growing interest in self-propelled systems in various forms, such as active colloids or autonomous robots, all more or less inspired by biological and social systems. Because they are out of equilibrium, novel theories must be developed to deal with such systems. This requires considerable effort to understand them fundamentally. The dependence of the steady-state particle density on their dynamical properties, as discussed by the authors, is a rather central prediction in this context, and it is definitely worth conducting extensive experimental testing.

Such a test is the main objective of the paper. As motile organisms, the authors use a mutant of E-Coli (developed in the Liphard Lab), which shows a rather immediate velocity response to changing conditions of light illumination. If the tested equation is true, these velocity changes should be reflected in their local density distribution. The authors provide experimental data for different patterns of light which partially agrees with theoretical predictions.

I think this is a well written and interesting paper which is certainly applicable for Nat. Comm. (despite quite some overlap with their previous Nat. Comm. publication). However, there are a couple of issues which should be addressed before I can support publication.

1. What is the reason for a circular E. coli motion at low concentrations?

This motion results from the hydrodynamic interactions of swimming bacteria close to a solid boundary and is explained in detailed in the cited reference [28]. As this effect is specific to the propulsion mechanism of flagellated E. coli, it might indeed not be widely known within the wider community and we have therefore added a few words of explanation to our manuscript (page 5).

2. The authors rationalize the deviations of the red symbols in Fig. 2a from a constant by saying that beyond $x = \pm 200\mu\text{m}$ the steady state could not be reached. But I am less convinced about the data in Fig. 3a where an almost constant downward slope of the red symbols is observed over the entire range. Then the agreement with a horizontal confidence interval is not really convincing.

We agree with the reviewer that the low density data shown in fig. 3a is far less convincing than the higher density data of fig. 2a. Indeed, here we cannot be certain that the remaining systematic slope in the central region close to the boundary is solely due to the system not having reached steady state. But the time evolution shown in fig 3b once again suggests that the system is still evolving, and the data close to the boundary is certainly also consistent with $\rho_s \cdot v = \text{const}$ within our confidence interval (which at $\sim 5\%$ is rather tightly defined!). We chose to show this data to highlight the experimental challenges at low density and how some of these could be overcome testing this relation only close to the step (fig. 4) or more convincingly by using the striped pattern shown in fig. 5.

3. The authors also used stripe patterns (Fig.5) with widths of $270\mu\text{m}$. This is about 3 times the persistent length of the bacteria with the knocked out cheY gene. Is this really sufficient to reach a steady state? I would like to see some further evidence for this, as this is a crucial requirement for the tested relation?

From the time course of our experimental data it is evident that the systems gets close to steady state within about 2-3 minutes of applying the striped pattern. As bacteria can enter the stripes from both directions, the distance bacteria have to travel to sample the spatial variation in speed is indeed comparable to their persistence length. So each swimming bacterium will be able to sample the spatial variations within a few persistence times, leading to a rapid approach to a steady state. All data points presented in the figure 5 show averaged values extracted from $\sim 40\text{s}$ of data, during which most bacteria will have sampled at least one of the intensity steps ($\Delta x \approx \sqrt{4Dt} = 400\mu\text{m}$ for $D \approx 1000 \mu\text{m}^2/\text{s}$ and $t = 40\text{s}$). After 15min striped illumination each cell will have crossed boundaries many time and thus the sample will have reached steady state.

4. Finally, it would be helpful to indicate some further implications of the tested relationship between particle density and particle dynamics. What are the consequences? How does it contribute to our understanding of collective behavior?

We have added a few more comments regarding the general applicability of $\rho_s \cdot v = \text{const}$ and more importantly the novel analysis method presented here to our conclusions (page 12).

Reviewer #3:

In this article, the authors probe the theoretical (1D) prediction relating the spatial density of active particles to their spatially dependent speed. In analogy to their previous work published in Nature Communications and the work from Di Leonardo's group published in e-life, they use bacteria that swim with an intensity dependent speed under spatially variable illumination generated with a DMD.

There are three major shortcomings that prevent publication of this work in Nature Communications:

1. This effect is not surprising or new: the fact that density is inversely proportional to speed is known since the 90's and was, for example, reported by Schnitzer for chemotactic systems. Another way to see this, is that particles will concentrate where their diffusivity is lower, and this has been confirmed experimentally both with passive and active systems, e.g. with active baths.

We are working under a widely-accepted understanding of the nature of science, namely, that nothing is 'known' for certain until verified by experiments. That was why, for instance, the Nobel Prize was only given to Peter Higgs *after* the boson was found at CERN. Schnitzer's work was theoretical. In any case, his prediction was for non-interacting systems of active particles, and we work with interacting swimmers throughout. So it cannot be said that $p v = \text{constant}$ 'is known since the 90s'. Both of the other referees recognise this and commended us for demonstrating this effect experimentally. Indeed, as the first reviewer pointed out: 'The applicability of the [theoretical] results to the experiment is in no way obvious. This is in particular since the theory does not account for hydrodynamic interactions which are present in the real system.'

There are two problems with 'it has been confirmed experimentally with passive and active systems'. Our understanding is that the dependence of density on dynamics is simply forbidden in passive systems at thermal equilibrium, so we are not sure what this reviewer means here. On the other hand, as far as active systems are concerned, we know of no such previous experimental confirmation, and the referee does not give any reference, as is common good practice.

We surmise that s/he may be referring to a failed recent attempt to demonstrate $p v = \text{constant}$ by Di Leonardo's group. We have discussed this in our original manuscript. Those authors used a strain of bacteria in which stopping was not instantaneous when illumination ceased, creating memory effects (which they acknowledged). They found $\rho = k/v + c$, where k and c are constants, or $p v = k + c v$, with c being comparable to k/v . There is therefore no sense in which they have demonstrated $p v = \text{constant}$. In any case, their (failed) attempt was merely *en passant*, and rather peripheral to their (beautiful) demonstration of bacterial painting with grey scale. We made a focussed, and successful, attempt at showing that $p v = \text{constant}$.

2. This demonstration does not have the broad appeal and interest for a multidisciplinary community as claimed in the article. The same effect was assumed and exploited, at least qualitatively, by the authors in their previous work. Although, it's nice to see, in principle, an actual quantification of this effect, I'm not convinced of its broad interest and applicability, especially when

limited to a 1D case as here. Real-life problems are barely 1D but, at least, 2D. Moreover, these results have very limited applicability as they can't be used for example with wild-type phototactic microorganisms.

Again, the disagreement here is philosophical: does the success of work that assumes the correctness of a result constitute sufficient proof of the result's correctness? Let's turn to actual scientific practice. That much of modern mathematics assumes the Riemann hypothesis has not stopped its proof remaining a 'holy grail'. The existence of gravitational waves was assumed in the 1993 Nobel Prize work on binary pulsars; the Nobel committee lauded those who detected such waves explicitly by awarding the 2017 physics prize to them. The field of active matter has very few clean, general predictions to date. $p v = \text{constant}$ is one of those. Common scientific practice therefore suggests that explicit demonstration of this result should count as a significant advance; reviewers 1 and 2 completely agree on this point.

In any case, it is doubtful whether, *sensu stricto*, it can be said that $p v = \text{constant}$ is a result that has been 'assumed ... qualitatively' in recent work on bacterial painting. This result is a strict equality, and there is no such thing as a 'qualitative' equality in physics. Two quantities are either equal or they are not. The bacterial painting work assumes any relation between p and v such that one variable increasing necessitates the other one decreasing, e.g., $p^2 v^3 = \text{constant}$, or indeed $p = k/v + c$. It is therefore simply wrong to suggest that the earlier bacterial painting work either by our group or by Di Leonardo's group in any way obviates the need for experimental confirmation of $p v = \text{constant}$. Indeed, as the first reviewer points out: 'While some evidence of this result was already given in Ref 11 of the manuscript ... the current manuscript goes well beyond this.'

This reviewer suggests that our result is not of general interest. Perhaps s/he is confusing interest with applicability. Some advances are of intrinsic scientific interest to all without being directly applicable to most. The Higgs boson is an obvious example. We claim general interest because the collective behaviour of micro-swimmers pertains to how physics deals with active agents in general. Results in this new statistical mechanics, one of which our experiments test, will ultimately lead to a general theory of living systems. So, while $p v = \text{constant}$ is not by itself widely applicable, the advance that this represents is a first step along a path is undoubtedly of wide general interest.

This reviewer seems to think that the inapplicability of $p v = \text{constant}$ to the run-and-tumble wild type constitutes a defect in our work. Quite to the contrary, this is a significant discovery in two senses. First, we have shown that tumbling rate increases temporarily when decreasing green illumination; this is an interesting finding for microbiology, but it also must be taken into account in all future active matter experiments using light and *E. coli*. Secondly, and central to our aim of establishing a fundamental result, literature proofs of $p v = \text{constant}$ depend on the tumbling rate being spatially independent. Our finding that a spatially-dependent tumbling rate (as obtains in the wild type) leads to a breakdown of $p v = \text{constant}$ therefore substantially adds to the completeness of our work.

3. The data are not convincing or presenting strong evidence. In Fig. 3 and 5 the authors for example claim that the red line is constant within the experimental error. However, if one considers the surrounding trends, it is hard to believe how such a claim can be made with confidence: in Fig. 3 data could easily fall on a line and in Fig. 5 on a sinusoidal function. Finally, there is also the fact that, from a technical point of view, this work does not add much to what the authors have already published in their previous paper.

With Figure 3, we quote our reply to Reviewer 2 to make clear our motivation for including this data: “We agree with the reviewer that the low density data shown in fig. 3a is far less convincing than the higher density data of fig. 2a. Indeed, here we cannot be certain that the remaining systematic slope in the central region close to the boundary is solely due to the system not having reached steady state. But the time evolution shown in 3b once again suggests that the system is still evolving, and the data close to the boundary is certainly also consistent with $\rho_s \cdot v = \text{const}$ within our confidence interval (which at ~5% is rather tightly defined!). We chose to show this data to highlight the experimental challenges at low density and how some of these could be overcome testing this relation only close to the step (fig. 4) or more convincingly by using the striped pattern shown in fig. 5.”

Figure 5 is a different matter all together. The undulations in the data are all (except for 1 point) within the grey shaded region, which indicates the ± 1 standard deviation noise level of our data. Thus, to attempt to fit the ρv data in Figure 5 with ‘a sinusoidal function’ would be a most elementary mistake in data analysis – one should never fit more precisely than the error margins allow! Assuming that this reviewer is not so naïve, s/he must then be challenging our error estimate. For the ‘sinusoidal’ undulations to be real, our error margins must be very considerably smaller than what we claim. We do not think that anyone who knows about this kind of experiments will believe that such low errors should be possible. Irrespective of our reported error margins, then, it is intrinsically unlikely that it would be legitimate to fit a ‘sinusoidal function’ to what is patently noise in our data.

As to practical applicability, we never claimed any. A fundamental physics result is worth proving irrespective of immediate utility. Reviewers 1 and 2 both agree on this point. As to whether we have added anything ‘from a technical point of view’ to recent publications on bacterial painting, we should point out that the use of spatially-resolved differential dynamic microscopy represents a significant technical advance. We admit that we have not highlighted this point in our original submission; this is now rectified both in the abstract and in the main body of the paper.

Reviewer #1 (Remarks to the Author):

I am happy with the changes and recommend publication.

Reviewer #2 (Remarks to the Author):

The revised version has addressed most of my questions and the quality of the manuscript has substantially improved.

Although I am still not very happy with Fig.3, the authors have ruled out my concerns regarding potential sampling problems related to Fig.5. Since Fig.5 is to my opinion central for the paper because it provides the strongest evidence for the authors claims. Therefore I am happy to support publication of the revised manuscript in Nat. Comm.